# Allelopathic and auto-toxic effects of aqueous extracts of *Codonopsis pilosula* root secretions

**Daiyu Qiu**[1,2], **Xue Wang**[1,2‡], **Fengbin Xu**[1,2], **Qian Li**[1,2], **Fengying Ren**[3], **Kan Jiang** [1,2]*

**1** College of Agronomy, Gansu Agricultural University, Lanzhou, China, **2** Gansu Key Laboratory of Arid Habitat Crop Science, Lanzhou, China, **3** Junggar Banner Baixin Pharmacy, Ordos City, Inner Mongolia Autonomous Region, China

‡ XW co-first author on this work.
* jiangk@gsau.edu.cn

**Data Availability Statement:** The data presented in this study are available from figshare with the DOI: 10.6084/m9.figshare.27225150.(https://figshare.com/s/9912687ee14181bfc582).

## Abstract

Water-soluble constituents in the root exudates of *C. pilosula* exhibit both allelopathic and auto-toxic properties, which substantially impede its growth. To mitigate the constraints associated with the continuous cropping of *C. pilosula*, this study examined the impact of aqueous extracts from the root exudates at various developmental stages on seed germination and seedling growth. Additionally, we isolated and characterized significant auto-toxic allelochemicals. Our findings revealed that the inhibitory effects on seed germination, seedling growth, and the activity of antioxidant enzymes-collectively referred to as a comprehensive effect-intensified progressively with increasing concentrations of the aqueous extracts from *C. pilosula* root exudates. The most pronounced auto-toxic comprehensive effect was observed with extracts at a concentration of 50 mL during the reproductive phase, diminishing SOD and POD activities, and MDA levels in *C. pilosula* seedlings by 72.3%, 71.7%, and 83.3% respectively, compared to the control ($P<0.05$). Three primary allelopathic compounds-acetic acid, hexamethyl cyclotrisiloxane, and methoxybenzene oxime were isolated and identified in the aqueous extracts. Acetic acid, constituting 44.1% of the total chemical profile, exhibited a significant inhibitory effect on seed germination and seedling growth ($P<0.05$). In conclusion, the aqueous extracts of *C. pilosula* root exudates significantly inhibit seed germination and seedling growth, especially during the plant's reproductive stage, with low molecular weight organic acids being the predominant allelopathic components.

## 1. Introduction

*Codonopsis pilosula*, a commonly utilized bulk medicinal material in China, as specified in the 2015 edition of the Chinese Pharmacopoeia, originates from three varieties of plants belonging to the Campanulaceae family, namely *Codonopsis pliosula* (Franch.) Nannf, *Codoncpsis pilosula* Nannf. var.*modesta* (Nannf) L.T.Shen, or *Codonopisis* tangshen Oliv [1]. The dry roots of these species possess the functions of tonifying the middle qi, strengthening the spleen, and benefiting the lungs. As the medicinal value of *Codonopsis pilosula* is progressively acknowledged by people, the market demand escalates year by year. The annual cultivation and total

**Funding:** This study was funded by the National Natural Science Foundation of China (No. 31960395). The funder provided financial support for the research activities, including the procurement of necessary materials and equipment.

**Competing interests:** The author has declared that there is no competing interests

output account for more than 90 percent of the nation. The demand for *Codonopsis pilosula* far exceeds the market supply, resulting in its extensive cultivation in a continuous cropping system over a large expanse of land. Nevertheless, this approach encounters substantial challenges. Research has affirmed that continuous cropping cultivation of melons, fruits, vegetables, grain crops, flowers, tobacco, and medicinal plants can all generate continuous cropping obstacle phenomena to varying extents [2–5]. Therefore, a comprehensive understanding of the mechanism and determining effective strategies to mitigate the continuous cropping challenges are of crucial importance for increasing the yield of *Codonopsis pilosula* in major planting areas. The limitations of continuous cropping are multifaceted and are related to factors such as imbalance of soil nutrients, severe soil-borne pests and diseases, degradation of the soil micro-ecological environment, and autotoxicity caused by root exudates or decomposition of plant residues.

Root exudates, particularly organic acid exudates, are regarded as the main contributors to autotoxic effects and have received significant attention in the study of plant continuous cropping obstacles [6]. Our previous research has indicated that the water extracts of the stems, leaves, and roots of *Codonopsis pilosula*, as well as the extracts of continuous cropping rhizosphere soil, can significantly inhibit the germination and seedling growth of other crops such as *Angelica sinensis*, wheat, and corn, demonstrating a strong allelopathic effect. The accumulation of reactive oxygen species (ROS) induced by autotoxins at the root tip of *Codonopsis pilosula* is considered an important factor affecting plant toxicity [7]. Wang et al. studied the effects of water extracts of *Codonopsis pilosula* straw at different concentrations on the seed germination and seedling growth of three traditional Chinese medicines, *Astragalus membranaceus*, *Scutellaria baicalensis*, and *Bupleurum chinense*. The results showed that the straw water extracts significantly reduced the seed germination rate and germination index of the three traditional Chinese medicines, and the higher the concentration of the water extract, the greater the inhibitory effect [8]. Ren et al. studied the allelopathic effects of root aqueous extracts of *Codonopsis pilosula* at different growth stages on the seed germination and seedling growth of *Codonopsis pilosula*, Vicia faba L, *Astragalus membranaceus*, and *Angelica sinensis*. It was found that the extracts showed allelopathic inhibitory effects on the seed germination and seedling growth of the four crops, and the inhibitory effect increased with the increase in the concentration of root aqueous extracts [9]. However, specific autotoxic and allelopathic characteristics that contribute to the continuous cropping challenges of *Codonopsis pilosula* are still poorly understood. This study aims to analyze the autotoxic effects and allelopathic mechanisms of *Codonopsis pilosula* rhizosphere exudates on seed germination and seedling growth and to isolate and identify compounds with significant allelopathic impacts. These findings provide a theoretical basis for understanding the role of allelochemicals in the roots of *Codonopsis pilosula* in arid agricultural areas of Gansu Province and help formulate strategies to mitigate the limitations associated with the continuous cropping of *Codonopsis pilosula*.

## 2. Materials and methods

### 2.1 Study site

This experiment was conducted in Tanchang County, Longnan City, Gansu Province (E 104˚ 01', N33˚46', altitude 2390 m). The region falls within the Longnan temperate humid zone and experiences a temperate continental climate. It is characterized by an average annual temperature of 9.3˚C, an average annual precipitation of 583.9 mm, an average annual sunshine duration of 1986.5 hours, and an average annual frost-free period of 181 days. The soil at the study site exhibits a pH of 8.2 and contains 12.5 g/kg of organic matter, 15.7 mg/kg of nitrate nitrogen, 22.6 mg/kg of available phosphorus, and 178.1 mg/kg of available potassium. Previously,

the site was used for cultivating *Astragalus memeranaceus* (Fisch.) Bge. Var. *mongholicus* (Bge.) Hsiao.

Seeds and seedlings of *C. pilosula*. Nannf. var. *modesta* (Nannf.) L. T. Shen, a variant of *Codonopsis platycodon*, were acquired from Tanchang Traditional Chinese Medicinal Materials Trade City.

## 2.2 Collection and hydroponic culture of *C. pilosula* seedlings

The intact root systems of 100 well-grown and pest-free *C. pilosula* seedlings were collected from the test site, washed with deionized water to remove soil debris, then sterilized by immersion in 1% potassium permanganate solution for 15 minutes and then rinsed with deionized water two to three times. The purified root samples were immediately sent to the laboratory and the *C. pilosula* seedlings were further cultivated under controlled conditions.

Hydroponics was employed to cultivate *C. pilosula* seedlings [10]. The seedlings, after being cleaned and sterilized, were positioned on foam boards with uniform perforations and subsequently placed into hydroponic tanks. A total of five hydroponic tanks were used, each hosting twenty *C. pilosula* seedlings. The tanks were filled with a 1/2 strength Hoagland nutrient solution [11], composed of pure water, Concentrate A masterbatch (primarily containing calcium nitrate and potassium nitrate), Concentrate B masterbatch (primarily containing ammonium phosphate and magnesium sulphate), and Concentrate C masterbatch (primarily containing ferrous sulphate heptahydrate and disodium ethylene diamine tetra acetic acid) mixed in a ratio of 17:1:1:1. The Hoagland nutrient solution provides balanced nutrition for plant growth, including macro-elements (e.g. nitrogen, phosphorus, potassium, calcium, magnesium, etc.) and micro-elements (e.g. iron, etc.), which can satisfy the nutritional needs of different plants at different growth stages. The tanks were aerated and exposed to light for 12 hours daily. The nutrient solution was replaced every five days to ensure optimal plant growth.

## 2.3 Preparation of aqueous solution of root secretion of *Codonopsis pilosula*

Root secretions were collected at three critical stages of ginseng growth (May 30th, July 29th, and September 28th). Fifteen plants were selected at each time, cleaned the root surface and put into a beaker containing 1 liter of sterile deionized water, covered the beaker with a black cloth to protect it from light, and then the roots were removed after four hours. The 1-liter aqueous solution containing root secretion was concentrated to 50 mL, 100 mL, and 150 mL respectively, then stored in a refrigerator at 4°C as the treatment solutions for the seed germination experiments. The collection was repeated every three days, three times for each stage, and the aqueous solutions of root secretions from the same growth period were combined for analytical experiments [12].

## 2.4 Determination of indicators and methods

**2.4.1 Determination of seed germination index of *Codonopsis pilosula*.** Distilled water served as the control, and three different concentrations of root secretion aqueous leachate were used as treatment groups, resulting in a total of 12 treatments. Each treatment was replicated three times. For the experiment, fifty uniformly sized, intact, and undamaged *C. pilosula* seeds, sterilized by soaking in a 15% $NaClO_3$ solution for 10 minutes, were placed on a petri dish containing 5 mL of each treatment solution.

The petri dishes with seeds were then incubated in a thermostatic incubator at 25°C, with a light intensity of 4,000 $\mu mol \cdot m^{-2} \cdot s^{-1}$ and a 12-hour light cycle. To maintain optimal conditions for germination, water was periodically sprayed to keep the filter paper moist. The number of

germinated seeds in each petri dish was counted daily. A seed was considered germinated when the seed radicle broke through the seed coat and reached at least half the length of the seed itself. Germinated seeds were then removed from the petri dishes to ensure accurate daily counts [13].

$$\text{Germination rate} = \text{Number of germinated seeds} \div \text{number of tested seeds} \times 100\%$$

Germination strength = Number of seeds germinated on day 3 ÷ total number of seeds for testing × 100%

**2.4.2 Determination of growth indexes of *Codonopsis pilosula* seedlings.** The "small cup method" was employed [14] for the cultivation of seedlings. Specifically, *C. pilosula* seedlings that had been grown for 7 days were transferred to horticulture flasks containing the appropriate concentration of immersion solution, and then placed back into the incubator for continued development. On the 20th day of growth, various growth indicators—including stem length, root length, and total biomass—along with physiological indicators, were measured to assess the health and development of the *C. pilosula* seedlings.

Stem length was quantified using vernier calipers to measure the straightened rhizomes of *C. pilosula* seedlings. Measurements were taken from the clear demarcation line between the above-ground and underground parts up to the tip of the leaf blade, with results expressed in centimeters.

Root length of *C. pilosula* seedlings was measured using vernier calipers. The measurement was taken from the clear demarcation line between the above-ground and underground parts of the straightened roots to the tip of the root, with the results expressed in centimeters.

Total fresh weight was determined by measuring the mass of seedlings in three petri dishes for each treatment.

**2.4.3 Verification of allelopathic effects of monomer compounds.** The most abundant monomer compounds were identified and prepared into solutions at concentrations of 1, 0.1, and 0.01 g/L, forming three concentration treatments. Each treatment was replicated three times, alongside a control group. For each replication, 50 uniformly sized *C. pilosula* seeds were randomly selected. These seeds underwent seed germination and seedling growth tests, using the same methods as previously described in sections 2.4.1 and 2.4.2, to evaluate the auto toxic effects of the monomer compounds.

The Williamson allelopathic response index ($I_r$) was utilized to quantify the type and intensity of allelopathic effects. The index is calculated using the formula $I_r = 1 - CK \div T$ (when $T \geq CK$) or $I_r = T \div CK - 1$ (when $T < CK$) [15], where CK is the mean of the control value indicators and T is the mean of the treatment group indicators; $I_r < 0$ indicates inhibition of seed germination and seedling growth and $I_r > 0$ indicates promotion of seed germination and seedling growth, and the magnitude of the absolute value is consistent with the strength of promotion or inhibition.

The allelopathic comprehensive effect index ($M_r$) is calculated as the arithmetic mean of the percentage of inhibition or promotion across multiple measured indicators of *C. pilosula* seedlings within the same treatment. These indicators typically include germination rate, germination strength, stem length, root length, and fresh weight. The formula to calculate Mr. is as follows:

$$M_r = \left( \sum_{j=1}^{n} I_j \right) \div n$$

where $M_r$ is the chemosensory composite effect index, $I$ is the $I_r$ value, and $n$ is the total number of indicators measured. $M_r > 0$ indicates a facilitating effect and $M_r < 0$ indicates an inhibitory effect.

**2.4.4 Determination of antioxidant enzyme activity.** Superoxide dismutase (SOD) activity was assessed using the nitrogen blue tetrazolium (NBT) photoreduction method. Peroxidase (POD) activity was measured utilizing the guaiacol method. Additionally, malondialdehyde (MDA) content was quantified through the two-component spectrophotometric method [16].

## 2.5 Identification of allelopathic and auto toxic substances

Gas chromatography-mass spectrometry (GC-MS) analysis was employed to qualitatively identify the chemical constituents in an aqueous solution of *C. pilosula* root secretion at a concentration of 125 mg/mL [17]. The analysis was conducted using an Agilent 7890B-7000D triple quadrupole gas chromatography-mass spectrometry system (Agilent, USA), equipped with an HP-5MS capillary column (30 m × 0.25 mm × 0.25 μm). The detection conditions were set as follows: a 3 μL injection volume, an injection temperature of 250˚C, and a temperature program that started at 50˚C for 2 minutes before ramping up to 210˚C at a rate of 5˚C/min for 10 minutes. Mass spectrometry settings included an electron ionization energy of 70 eV, a scanning range of 35–600 m/z, an ion source temperature of 230˚C, and a quadrupole temperature of 150˚C. Helium was used as the carrier gas at a flow rate of 1 mL/min. The mass spectra were analyzed using the NIST mass spectrometry database for the identification of compounds, and the peak area normalization method was applied to calculate the relative content of each component.

## 2.6 Data processing

We used SPSS 26.0 (IBM SPSS Inc., Chicago, USA) software for one-way analysis of variance (ANOVA) with a significance level of $\alpha = 0.05$. Correlation analyses and graphing were performed using Origin 2021 (www. Origin Lab. com) software.

## 3. Results

### 3.1 Effects of aqueous extracts of root secretions of *C. pilosula* on own seed germination and seedling growth at different growth stages

The chemosensitization indices of different concentrations of the leaching solution were all negative (Ir $<$ 0), indicating that the leaching solution of the root secretion of *Codonopsis pilosula* inhibited seed germination and seedling growth, and the inhibitory effect of the leaching solution of the root secretion of *Codonopsis pilosula* on the germination rate and germination potential of *Codonopsis pilosula* seeds in different reproductive periods showed the trend of "weak at the early stage, strengthened in the reproductive stage, and weakened in the harvesting stage". "At the same time, the inhibitory effect on seed germination and seedling growth of *Codonopsis pilosula* increased with the concentration of the extract. The germination rate |Ir| value$<$ germination potential |Ir|value of *Codonopsis pilosula* seeds at different concentrations in each reproductive period indicated that the aqueous extract of root secretion of *Codonopsis pilosula* had a greater degree of influence on the germination potential of its seeds than the germination rate (Fig 1A and 1B). The stem length, root length, and total fresh weight of *Codonopsis pilosula* seedlings decreased continuously with the concentration of the aqueous solution. The aqueous solution concentrated to 50 mL had the most significant inhibitory effect on the stem length, root length, and total fresh weight of *Codonopsis pilosula* (Fig 1C–1E), which indicated that the concentration of root exudates is positively correlated with the degree of inhibition of *Codonopsis pilosula* seedlings. In actual planting, it was necessary to consider adopting certain methods to reduce the accumulation of root exudates to prevent them from inhibiting the growth of *Codonopsis pilosula*. Comprehensively analyzing each growth index, the chemosensory comprehensive effect index was also negative, all of which

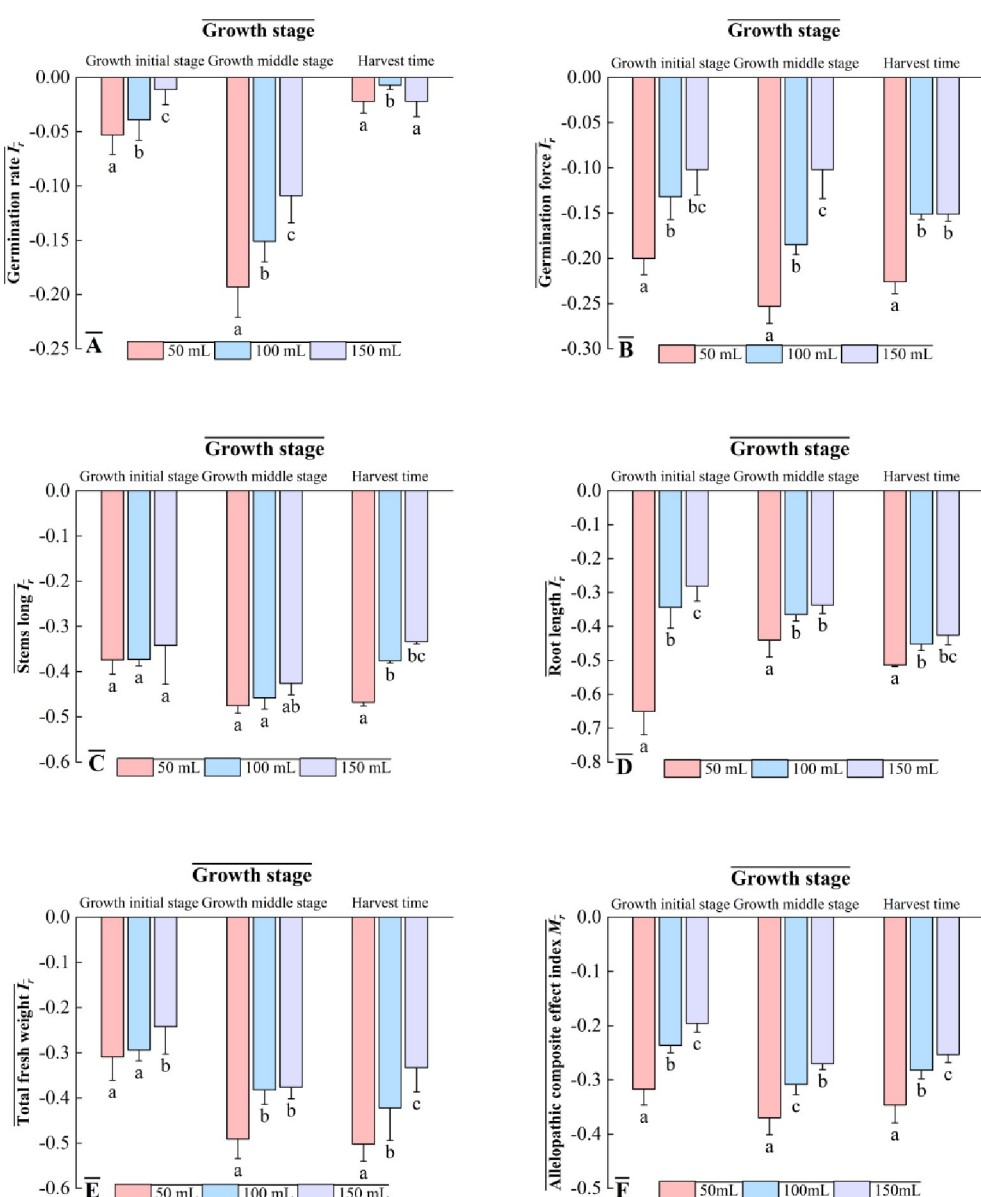

**Fig 1. Allelopathy index (Ir) of *C. pilosula* seed germination and seedling growth under the treatment of aqueous extracts of root secretion of *C. pilosula*.**

were chemosensory inhibition (Mr < 0), indicating that the autotoxic effect of *Codonopsis pilosula* increased with the increase of the concentration of the aqueous solution, and the inhibition of the aqueous solution of root secretion, which was in the reproductive stage of *Codonopsis pilosula* and concentrated up to 50 mL, was the most significant, with the strongest autotoxic comprehensive effect (Fig 1F).

## 3.2 Effects of aqueous extracts of root secretions of *C. pilosula* on antioxidant enzyme activities own seedlings at different growth stages

Different concentrations of aqueous extracts of *C. pilosula* root secretions significantly reduced the activity of reactive oxygen scavengers and led to a significant accumulation of lipid

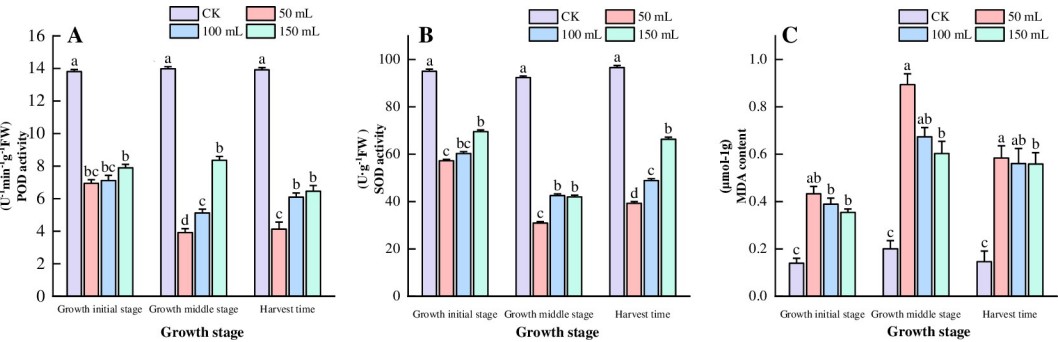

**Fig 2. Effects of aqueous extracts of root exudates of *C. pilosula* on antioxidant enzyme activity and malondialdehyde content in *C. pilosula* seedlings.**

peroxidation products in *C. pilosula* seedlings at different reproductive periods (Fig 2). Both superoxide dismutase (SOD) and peroxidase (POD) activities gradually decreased with increasing concentrations of aqueous extracts of *C. pilosula* root secretions as plant growth progressed. The aqueous extracts concentrated to 50 mL exhibited the most significant reduction in POD activity across different plant growth periods, with the weakest POD activity observed during the reproductive stage, which was 71.7% lower than the control (Fig 2A). During the propagation period of *C. pilosula*, different concentrations of root secretions reduced SOD activity by 55.8% to 72.3% compared with the control. The weakest SOD activity was observed under the 50 mL concentration of root secretion aqueous leachate (Fig 2B). At different growth stages, 50 mL of the concentrate resulted in the highest accumulation of malondialdehyde (MDA) content in the leaves of *C. pilosula* seedlings, significantly different from the control (P < 0.05). The MDA content increased by 83.3% during the reproductive period with the 50 mL concentrate compared to the control (Fig 2C).This indicated that the aqueous extract of *Codonopsis pilosula* root exudates could inhibit the antioxidant system of *Codonopsis pilosula* seedlings, resulting in increased accumulation of lipid peroxidation products.

## 3.3 Identification of allelopathic components in the aqueous extract of root exudates of *C. pilosula*

The five main components in the root secretions of C. *pilosula* were acetic acid ($C_2H_4O_2$), hexamethyl cyclotrisiloxane ($C_6H_{18}O_3Si_3$), methoxybenzene oxime ($C_8H_9NO_2$), L-Tyrosine 3-carboxy- ($C_{10}H_{11}NO_5$), and benzene 1-benzyloxy-5-diethylamino-2,4-dinitro- ($C_{17}H_{19}N_3O_5$). Among them, the relative content of acetic acid was the highest, accounting for 44.1% of all chemical constituents. The relative contents of hexamethyl cyclotrisiloxane and methoxybenzene oxime were the next highest, each accounting for 22.3% of all chemical constituents. The remaining components were each less than 10% (Fig 3). The three components with the highest content exhibited allelopathic effects [18] (Fig 4).

## 3.4 Validation of allelopathic effects of different concentrations of acetic acid

Acetic acid had an inhibitory effect on both seed germination and seedling growth of *C. pilosula*, with a significant concentration gradient effect (P < 0.05, Table 1).

The germination rate and germination strength of *C. pilosula* seeds, along with the stem length, root length, and total fresh weight of *C. pilosula* seedlings, decreased significantly with increasing concentrations of acetic acid. The allelopathy index was -1 at 1 g/L acetic acid,

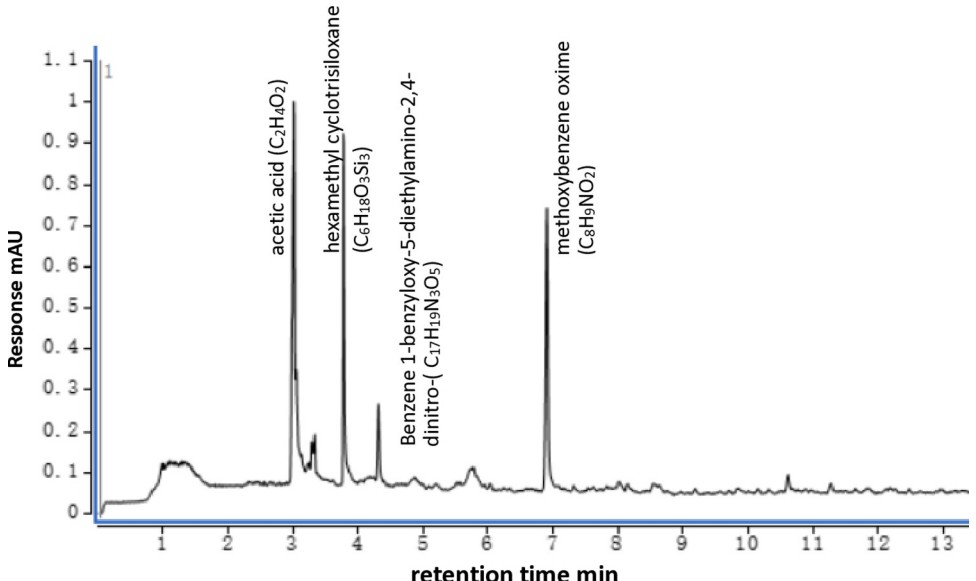

**Fig 3. Total ion flow chromatogram of root exudates of *C. pilosula*.**

indicating a 100% inhibition rate. Higher concentrations of acetic acid resulted in a stronger inhibitory effect on the POD and SOD activities of *C. pilosula* seedling leaves. At 1 g/L, the inhibition rate on POD and SOD activities reached 100%, indicating that this concentration would completely damage the protective enzyme system of *C. pilosula* leaves and cause the plants to lose their antioxidant capacity.

The MDA content in *C. pilosula* seedlings accumulated under both 0.01 g/L and 0.1 g/L acetic acid treatments, reaching the highest level at a concentration of 1 g/L. This suggested that increasing acetic acid concentrations could enhance membrane lipid peroxidation in the tissues or organs of *C. pilosula* seedlings, leading to greater damage to the cell membranes of the leaves.

## 4. Discussion and conclusions

The germination of seeds and the growth of seedlings can be employed to assess the auto-toxic effects of allelochemicals on plants. Allelochemicals produced by root exudates can inhibit seed germination, disrupt the growth processes of plants, and directly or indirectly affect the growth and development of adjacent plants of the same or subsequent species [19]. In this

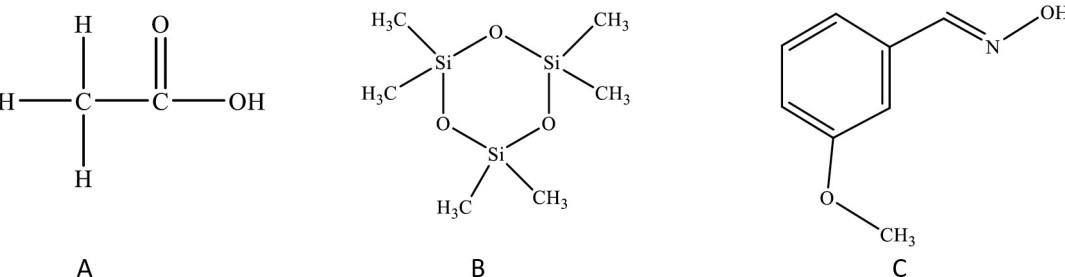

**Fig 4. The three most Abundant Allelochemical Components in the Aqueous Extracts of *C. pilosula* Root Exudates: A: Acetic Acid ($C_2H_4O_2$), B:Hexamethyl cyclotrisiloxane ($C_6H_{18}O_3Si_3$), C:Methoxybenzene Oxime ($C_8H_9NO_2$).**

**Table 1. The allelopathic index ($I_r$) of *C. pilosula* seed germination and seedling growth under different concentrations of acetic acid.**

| Index | Acetic acid concentration | | |
|---|---|---|---|
| | 1 g/L | 0.1 g/L | 0.01 g/L |
| Germination rate | -1.000±0.031a | -0.197±0.015b | -0.109±0.011c |
| Germination strength | -1.000±0.032a | -0.623±0.009b | -0.645±0.016b |
| Stem length | -1.000±0.021a | -0.616±0.021b | -0.595±0.013bc |
| Root length | -1.000±0.013a | -0.571±0.023b | -0.287±0.025c |
| Fresh weight | -1.000±0.031a | -0.670±0.022b | -0.556±0.014bc |
| POD (U/min/g/FW) | -1.000±0.022a | -0.512±0.026b | -0.491±0.019bc |
| SOD (U·g/FW) | -1.000±0.011a | -0.396±0.025b | -0.251±0.012bc |
| MDA (μmol/g) | -1.000±0.042a | -0.794±0.023b | -0.741±0.017bc |

Note: Different lowercase letters indicate significant differences among concentrations at the $P<0.05$ level.

study, the aqueous extracts of root exudates of *Codonopsis pilosula* at different growth stages have a significant inhibitory effect on the seed germination and seedling growth of *Codonopsis pilosula*, and the auto-toxic effect is most significant during the reproductive stage among them. This might be due to the changes in plant metabolic pathways and the accumulation of decomposition products of root exudates in the soil as the plant grows and develops, which aggravates the auto-toxic effect of *C. pilosula* at the reproductive stage. Yu et al. found that the exudates at the reproductive stage are more phytotoxic than those at the vegetative stage [20]. The research results also verified that the allelopathic auto-toxic substances exhibit an obvious dose-dependent effect on the growth of *C. pilosula*. This result is consistent with the research results of Zhou [21] and is also in line with the research results of the auto-toxic effects of root extracts and stem-leaf extracts of *C. pilosula* by our team [22, 23].

Malondialdehyde (MDA), as the main product of membrane lipid peroxidation in plants, its content can reflect the degree of damage to the biological membrane system. By determining the content of MDA, the damaged condition of the plant's biological membrane system can be inferred [24]. Antioxidant enzymes (such as SOD, POD, and CAT, etc.) play an important role in plants. They are the key substances for preventing membrane lipid peroxidation, and can effectively scavenge reactive oxygen species in plants and reduce peroxidative damage [25]. The study found that the aqueous extract of *C. pilosula* root can induce the expression of antioxidant enzyme activities to different degrees, thereby regulating the content of malondialdehyde in *C. pilosula* seedlings. Compared with the control group, it significantly reduced the activities of these enzymes and promoted the content of MDA (Fig 2), which is basically consistent with the effects of the aqueous extracts of root exudates of Chinese medicinal materials such as *Angelica sinensis*, *Astragalus membranaceus*, and *Atractylodes macrocephala* on the growth of plant seedlings [26–28]. Different concentrations of the aqueous extract of *C. pilosula* root can cause allelopathic stress on seeds and seedlings. At different growth stages of *C. pilosula*, the allelopathic inhibitory effect of its root extract is different. The allelopathic inhibitory effect is the strongest at the reproductive stage, followed by the harvest stage, and the weakest at the early growth stage (Fig 2). In different growth seasons, there are differences in the types and contents of allelochemicals, which leads to different intensities of allelopathic effects on plants at different developmental stages [29]. In addition, the growth and metabolism of *C. pilosula* are active at the reproductive stage, and the types or quantities of allelochemicals produced are relatively large, thus showing a strong allelopathic auto-toxic effect.

The secretions released by the root system of *Codonopsis pilosula* to the rhizosphere during its growth are not only important carriers for the exchange of matter, energy, and information

between the plant and the soil environment but also significant sources of allelopathic substances [30]. In dry-farming areas, the water-soluble components of these secretions are transferred to the soil through rainwater leaching, thereby influencing the next crop of *Codonopsis pilosula*. Among the four major types of allelochemicals in plants (fatty acids, aliphatic, terpene, and aromatic compounds), the most common ones are low-molecular-weight organic acids, phenols, and end-terpene compounds. The main allelochemical substance isolated and identified from the rhizosphere soil of *Codonopsis pilosula* is Codonopilate A, and the compounds with weak auto-toxicity include taraxasteryl acetate and 24-methylenecycloartanol [7, 31]. Low-concentration phenolic acids can induce the enhancement of the activity of the plant's protective enzyme system to resist stress. However, as the concentration of phenolic acids increases, the induced damage exceeds the plant's self-protective capacity, instead inhibiting the protective enzymes and leading to the accumulation of harmful substances such as MDA [32]. As a common allelochemical, acetic acid is significantly harmful to plants. Firstly, acetic acid can interact with the components of the cell membrane, changing its physical and chemical properties, resulting in increased membrane permeability and the leakage of intracellular substances. This not only disrupts the intracellular material balance and interferes with the cell metabolic pathways but also damages the cell structure, affects the function of organelles, and hinders the cell's absorption of water and nutrients. Secondly, acetic acid can disrupt cell metabolism, leading to the accumulation of reactive oxygen species (ROS) [33]. These ROS can attack the unsaturated fatty acids in the cell membrane, triggering lipid peroxidation. The products of lipid peroxidation, such as malondialdehyde (MDA), can cross-link with membrane proteins and phospholipids, further damaging the structure and function of the membrane. This kind of damage not only affects the selective permeability of the membrane but also disrupts the normal operation of ion channels and transport proteins, thus interfering with the absorption and transport of water and nutrients. In addition, the loss of root ions caused by acetic acid can break the ion balance within the plant, interfere with many metabolic processes, and reduce the plant's stress-resistance ability [34]. Consistent with the results of this study, as the concentration of acetic acid increases, the accumulation of MDA content also gradually increases (Table 1), which leads to lipid peroxidation of the plant cell membrane and damages the structure and function of the cell membrane. In conclusion, there are low-molecular-weight organic acid auto-toxic substances in the water extract of *Codonopsis pilosula* roots, and their allelopathic auto-toxic effect is significant, which is one of the incentives for the continuous cropping obstacles of *Codonopsis pilosula*. The enrichment of allelochemicals and their auto-toxicity not only affects the seed germination, seedling growth, and the activity of related enzymes of *Codonopsis pilosula* but also runs through the entire process of its growth and development. Therefore, the isolation and identification of allelochemicals of *Codonopsis pilosula* and the exploration of their auto-toxic mechanism from multiple perspectives are the keys to solving the problem of continuous cropping obstacles of *Codonopsis pilosula*.

In the study of the aqueous extract of *Codonopsis pilosula* root secretions, we have found that some important factors have been overlooked, such as volatile compounds, the interaction between soil and root-secreted substances, and microbial activity. Future research will focus on exploring the role of allelochemicals in these aspects, such as studying the influence of volatile compounds on allelopathic effects, the change of allelochemical activity by soil-root secretion interaction, and the role of microbial activity in the allelopathic process. By addressing these issues, we hope to understand the allelopathic mechanism of *Codonopsis pilosula* root secretions more comprehensively. The study of the allelopathic auto-toxicity of *Codonopsis pilosula* has implications for agricultural practice. Understanding allelopathy can guide crop rotation and intercropping, improve soil fertility, and reduce pests. Moreover, the allelochemicals in root exudates affect the soil microbial community and fertility. We can improve soil

quality by regulating allelochemicals. In conclusion, this research result can promote the sustainable development of agriculture and maintain the health of the agricultural ecosystem.

## Acknowledgments

We would like to express our sincere gratitude to all the authors for their concerted efforts and hard work during the research and manuscript preparation. Notably, Wang Xue has been listed as a co-first author in recognition of her outstanding contributions to this work.

## Author Contributions

**Conceptualization:** Daiyu Qiu.

**Data curation:** Xue Wang, Fengying Ren.

**Formal analysis:** Xue Wang, Qian Li, Fengying Ren.

**Funding acquisition:** Daiyu Qiu.

**Investigation:** Xue Wang, Fengying Ren.

**Methodology:** Xue Wang, Fengbin Xu.

**Project administration:** Qian Li.

**Resources:** Fengbin Xu, Fengying Ren.

**Software:** Fengbin Xu, Kan Jiang.

**Supervision:** Qian Li, Kan Jiang.

**Validation:** Fengbin Xu.

**Visualization:** Kan Jiang.

**Writing – original draft:** Daiyu Qiu.

**Writing – review & editing:** Daiyu Qiu, Kan Jiang.

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
