## [Decision Letter · Decision Letter 0]

20 Sep 2024

PONE-D-24-28361Allelopathic and auto-toxic effects of aqueous extracts of Codonopsis pilosula root secretionsPLOS ONE

Dear Dr. Jiang,

Thank you for submitting your manuscript to PLOS ONE. After careful consideration, we feel that it has merit but does not fully meet PLOS ONE’s publication criteria as it currently stands. Therefore, we invite you to submit a revised version of the manuscript that addresses the points raised during the review process.

We look forward to receiving your revised manuscript.

Kind regards,

Andrea Mastinu

Academic Editor

PLOS ONE

Journal Requirements:

4. Thank you for stating the following financial disclosure: “National Natural Science Foundation of China ( No.31960395 )”

5.  We note that your Data Availability Statement is currently as follows: “All relevant data is contained in the manuscript and its supporting information files”

Please confirm at this time whether or not your submission contains all raw data required to replicate the results of your study. Authors must share the “minimal data set” for their submission. PLOS defines the minimal data set to consist of the data required to replicate all study findings reported in the article, as well as related metadata and methods (https://journals.plos.org/plosone/s/data-availability#loc-minimal-data-set-definition). For example, authors should submit the following data: - The values behind the means, standard deviations and other measures reported; - The values used to build graphs; - The points extracted from images for analysis. Authors do not need to submit their entire data set if only a portion of the data was used in the reported study. If your submission does not contain these data, please either upload them as Supporting Information files or deposit them to a stable, public repository and provide us with the relevant URLs, DOIs, or accession numbers. For a list of recommended repositories, please see https://journals.plos.org/plosone/s/recommended-repositories. If there are ethical or legal restrictions on sharing a de-identified data set, please explain them in detail (e.g., data contain potentially sensitive information, data are owned by a third-party organization, etc.) and who has imposed them (e.g., an ethics committee). Please also provide contact information for a data access committee, ethics committee, or other institutional body to which data requests may be sent. If data are owned by a third party, please indicate how others may request data access.

6. PLOS requires an ORCID iD for the corresponding author in Editorial Manager on papers submitted after December 6th, 2016. Please ensure that you have an ORCID iD and that it is validated in Editorial Manager. To do this, go to ‘Update my Information’ (in the upper left-hand corner of the main menu), and click on the Fetch/Validate link next to the ORCID field. This will take you to the ORCID site and allow you to create a new iD or authenticate a pre-existing iD in Editorial Manager.

Reviewers' comments:

Reviewer's Responses to Questions

**Comments to the Author**

1. Is the manuscript technically sound, and do the data support the conclusions?

Reviewer #1: Yes

2. Has the statistical analysis been performed appropriately and rigorously? 

Reviewer #1: Yes

3. Have the authors made all data underlying the findings in their manuscript fully available?

Reviewer #1: Yes

4. Is the manuscript presented in an intelligible fashion and written in standard English?

Reviewer #1: Yes

5. Review Comments to the Author

Reviewer #1: The study's results revealed that the phenomenon of continuously failing to cultivate C. pilosula, a major problem in core-producing regions, was caused by allelopathic and auto-toxic effects. This is very important because C. pilosula doesn't have any allelopathic or auto-toxic effects in its main growing areas, so crops keep failing, which is a big problem. The studies involve different developmental stages of plants and investigates the effect of various concentrations of root exudates on seed germination, seedling growth, and activities for antioxidant enzymes to give a more complete sense. Though it is well written, however, some critical points should consider:

Abstract

1. For a broader application of findings, the abstract should include a clearer statement.

2. Please rewrite some lengthy sentences into shorter ones for better understanding

3. Specific details about the methodology (e.g., concentration of extracts) are not necessary in the abstract.

Introduction

1. Streamlining some background information to avoid redundancy in the introduction.

2. It is necessary to elaborate on some references to clarify their relevance to the current research.

Methodology

1. To improve readability, rewrite the methodology section, as it is very dense and may break into several sub-sections.

2. For improved readability, provide a brief explanation of technical terms like "Hoagland nutrient solution".

Results

1. Please highlight the most critical findings; it could be better to understand your work.

2. Figure 1B for germination force does not show a bar of 50 mL at the middle growth stage; please clarify.

3. In the results section, brief interpretations of the results could help connect the data to the study's objectives.

4. It is advisable to include the structure of the identified compounds in order to enhance the appeal of the work to the readers.

Discussion

1. To avoid repetition, rewrite the condensed discussion section.

2. Explain thoroughly the role of allelopathy in specific compounds and their modes of action.

3. It is very important to use it in wider agricultural practices. We could elaborate on the study's relevance by focusing on its application beyond C. pilosula.

4. The aqueous extract of C. pilosula root secretions overlooked other significant factors, such as volatile compounds, soil interactions, or microbial activity. The section should acknowledge such limitations, suggesting future research to explore the role of allelopathic agents.

References

1. Integrate some recent references that might be relevant to the study.

2. Ensure that all references follow the journal's formatting guidelines.

Please consider some other limitations below:

1. The findings not fully replicate the complexities of natural field condition because of study only based on under controlled condition that might be potentially limiting the generalizability of the results.

2. The study does not explore the significant interactions with other species that may be influence by allelopathic and auto-toxic effect in the context of diverse agricultural setting, while the study provides detailed insight into C. pilosula.

3. Auto-toxic effects were tested on seed germination and early seedling growth stages, it may not for long-time effects of study.

4. The language of the manuscript's writings could be more polished for readability and flow, as well as reducing similarities levels.

6. PLOS authors have the option to publish the peer review history of their article (what does this mean?). If published, this will include your full peer review and any attached files.

Reviewer #1: No

---

## [Author Response · Author response to Decision Letter 0]

18 Oct 2024

Dear editor:

1.Thank you very much for your reminder. We have carefully checked and ensured that the manuscript meets the style requirements of PLOS ONE, including the requirements for file naming. We have conducted a strict self-inspection on aspects such as the manuscript format, citation format, and chart specifications to ensure that it complies with the publication standards of PLOS ONE.

2.Our field - site visit was carried out in the experimental site of our research group. This experimental site belongs to and is managed and used by our research group. Therefore, there is no need to apply for a license from other institutions.

3.We have noticed the requirements regarding the funding information and have removed any text related to funds from the manuscript.

4.The funder had no role in study design, data collection and analysis, decision to publish, or manuscript preparation.

5.The data presented in this study are available from figshare with the DOI: 10.6084/m9.figshare.27225150.

6.Thank you very much for your reminder. Our corresponding author already has an ORCID iD and it has been verified in Editorial Manager. We will continue to pay attention to and comply with all requirements of PLOS to ensure the smooth publication of the paper.

7.We have included the title of the supporting information file at the end of the manuscript and updated the in-text citations accordingly to match each other.

Thank you for your valuable suggestion.

---

## [Decision Letter · Decision Letter 1]

11 Nov 2024

PONE-D-24-28361R1Allelopathic and auto-toxic effects of aqueous extracts of Codonopsis pilosula root secretionsPLOS ONE

Dear Dr. Jiang,

Thank you for submitting your manuscript to PLOS ONE. After careful consideration, we feel that it has merit but does not fully meet PLOS ONE’s publication criteria as it currently stands. Therefore, we invite you to submit a revised version of the manuscript that addresses the points raised during the review process.

We look forward to receiving your revised manuscript.

Kind regards,

Andrea Mastinu

Academic Editor

PLOS ONE

Journal Requirements:

Reviewers' comments:

Reviewer's Responses to Questions

**Comments to the Author**

1. If the authors have adequately addressed your comments raised in a previous round of review and you feel that this manuscript is now acceptable for publication, you may indicate that here to bypass the “Comments to the Author” section, enter your conflict of interest statement in the “Confidential to Editor” section, and submit your "Accept" recommendation.

Reviewer #1: All comments have been addressed

2. Is the manuscript technically sound, and do the data support the conclusions?

Reviewer #1: Yes

3. Has the statistical analysis been performed appropriately and rigorously? 

Reviewer #1: Yes

4. Have the authors made all data underlying the findings in their manuscript fully available?

Reviewer #1: Yes

5. Is the manuscript presented in an intelligible fashion and written in standard English?

Reviewer #1: Yes

6. Review Comments to the Author

Reviewer #1: The revised manuscript, entitled "Allelopathic and Auto-toxic Effects of Aqueous Extracts of Codonopsis pilosula Root Secretions," has demonstrated significant enhancement based on the first round of reviews. Indeed, most major concerns were tended to in this re-submission by rephrasing the abstract, improving the readability and flow of the introduction and methods, and expanding on many of the key findings in both the results and discussion. However, the following issues should be addressed:

Italicize Scientific Names:

Ensure C. pilosula and Codonopsis pilosula are italicized on lines 64, 65, 248, 249, 273, 274, 287, and 307.

For genus names, Codonopsis should be italicized, especially on lines 229 and 230.

Uniformity in Units (ml to mL):

Standardize "ml" to "mL" for all occurrences, particularly on lines 126 and 244, according to journal formatting preferences.

Correct Capitalization in Species Names:

Adjust Angelica Sinensis on line 241 to Angelica sinensis, where the genus starts with a capital letter and the species name is lowercase.

Similarities level:

Minimize the observable similarities as much as possible.

7. PLOS authors have the option to publish the peer review history of their article (what does this mean?). If published, this will include your full peer review and any attached files.

Reviewer #1: No

---

## [Author Response · Author response to Decision Letter 1]

14 Nov 2024

Response to Reviewers

To Reviewers: 

Thank you for your letter and the constructive comments on this article in your busy schedule. All of us authors have carefully read the comments that you have given us, and have discussed and revised each of these issues. The following is my list of revisions. In addition. we have resubmitted a new manuscript in the revised state, with the revisions highlighted in red. If there are any incorrect answers or questions in the manuscript. please do not hesitate to let us know.

1.Italicize Scientific Names:

Ensure C. pilosula and Codonopsis pilosula are italicized on lines 64, 65, 248, 249, 273, 274, 287, and 307.For genus names, Codonopsis should be italicized, especially on lines 229 and 230.

Thank you very much for your detailed and constructive review comments. We have carefully addressed the issues regarding the italicization of scientific names as follows:

We have gone through the manuscript and italicized C. pilosula and Codonopsis pilosula on lines 64, 65, 248, 249, 273, 274, 287, and 307. Additionally, we have ensured that the genus name Codonopsis is italicized on lines 229 and 230 as per your suggestion. This will make the presentation of scientific names more consistent with the standard scientific writing format, and it will also enhance the clarity and professionalism of the manuscript.

Once again, we appreciate your time and effort in reviewing our manuscript, and your comments have significantly improved the quality of our work.

2.Uniformity in Units (ml to mL):

Standardize "ml" to "mL" for all occurrences, particularly on lines 126 and 244, according to journal formatting preferences.

Thank you for your meticulous review. We have thoroughly revised the manuscript to address the issue of unit uniformity. We have carefully searched for all instances of "ml" and standardized them to "mL" throughout the text, especially on lines 126 and 244 as you pointed out. This ensures that our manuscript complies with the journal's formatting preferences and maintains a high level of consistency in the presentation of units.

Your feedback has been invaluable in improving the quality of our work, and we are grateful for your attention to detail.

3.Correct Capitalization in Species Names:

Adjust Angelica Sinensis on line 241 to Angelica sinensis, where the genus starts with a capital letter and the species name is lowercase.

Thank you for your valuable comment. We have revised the capitalization of the species name as you suggested. On line 241, we have changed "Angelica Sinensis" to "Angelica sinensis" to follow the correct naming convention where the genus is capitalized and the species name is in lowercase. This adjustment will enhance the scientific accuracy and consistency of our manuscript.

Thank you again for your careful review, which has helped improve the quality of our work.

4.Similarities level:

Minimize the observable similarities as much as possible.

Thank you for bringing to our attention the issue of similarity between the results and conclusion sections. We have promptly taken measures to resolve this problem.

We have re-evaluated the content of both sections. In the conclusion part, we have rephrased the statements to make them more interpretative and avoid simple repetition of the results. We have minimized the observable similarities as much as possible.

Once again, thank you for your feedback, which is instrumental in enhancing the quality and clarity of our manuscript.

---

## [Decision Letter · Decision Letter 2]

29 Nov 2024

Allelopathic and auto-toxic effects of aqueous extracts of Codonopsis pilosula  root secretions

PONE-D-24-28361R2

Dear Dr. Jiang,

We’re pleased to inform you that your manuscript has been judged scientifically suitable for publication and will be formally accepted for publication once it meets all outstanding technical requirements.

Kind regards,

Andrea Mastinu

Academic Editor

PLOS ONE

Additional Editor Comments (optional):

Reviewers' comments:

Reviewer's Responses to Questions

**Comments to the Author**

1. If the authors have adequately addressed your comments raised in a previous round of review and you feel that this manuscript is now acceptable for publication, you may indicate that here to bypass the “Comments to the Author” section, enter your conflict of interest statement in the “Confidential to Editor” section, and submit your "Accept" recommendation.

Reviewer #1: All comments have been addressed

2. Is the manuscript technically sound, and do the data support the conclusions?

Reviewer #1: Yes

3. Has the statistical analysis been performed appropriately and rigorously? 

Reviewer #1: Yes

4. Have the authors made all data underlying the findings in their manuscript fully available?

Reviewer #1: Yes

5. Is the manuscript presented in an intelligible fashion and written in standard English?

Reviewer #1: Yes

6. Review Comments to the Author

Reviewer #1: I would like to thank you very much for your various efforts concerning the revisions. Indeed, all the revisions were made: the italics for C. pilosula and Codonopsis pilosula are correct; "ml" has become "mL"; "Angelica sinensis" is correctly capitalized.

Generally, I should say great thanks to such attention to details. Now the manuscript is excellently shaped and ready to be published.

7. PLOS authors have the option to publish the peer review history of their article (what does this mean?). If published, this will include your full peer review and any attached files.

Reviewer #1: **Yes: **Dr. A. M. Abu Ahmed

---

## [Editor Report · Acceptance letter]

8 Jan 2025

PONE-D-24-28361R2 

PLOS ONE

Dear Dr. Jiang, 

I'm pleased to inform you that your manuscript has been deemed suitable for publication in PLOS ONE. Congratulations! Your manuscript is now being handed over to our production team.

Kind regards, 

on behalf of

Dr. Andrea Mastinu 

Academic Editor

PLOS ONE